Development and application of multiplex PCR for the rapid identification of four Fusarium spp. associated with Fusarium crown rot in wheat

Deng Siyi 1 2 3
Chang Wei 1 2 3
Liu Quanke 4
Zhao Youfu 5
Liu Jun 1 2 3 liuj@hbaas.com
http://orcid.org/0000-0003-2552-7063 Wang Hua 1 2 3 wanghua4@163.com
1 Institute of Plant Protection and Soil Fertilizer, Hubei Academy of Agricultural Sciences , Wuhan , China
2 Hubei Key Laboratory of Crop Disease, Insect Pests and Weeds Control , Wuhan , China
3 Key Laboratory of Integrated Pest Management on Crops in Central China, Ministry of Agriculture , Wuhan , China
4 General Plant Protection Station of Hubei Province , Wuhan , China
5 Department of Plant Pathology, Irrigated Agriculture Research and Extension Center, Washington State University , Prosser , United States
Ribeiro-Barros Ana
Electronic publication date: 2024 Jun 27
Publication date: 2024
Volume: 12
Electronic Location ID: e17656
Received 2024 Apr 12; Accepted 2024 Jun 8
Copyright: © 2024 Deng et al.
Copyright year: 2024
Copyright holder: Deng et al.
License: This is an open access article distributed under the terms of the Creative Commons Attribution License, which permits unrestricted use, distribution, reproduction and adaptation in any medium and for any purpose provided that it is properly attributed. For attribution, the original author(s), title, publication source (PeerJ) and either DOI or URL of the article must be cited.
License URL: https://creativecommons.org/licenses/by/4.0/

Keywords: Fusarium crown rot, Fusarium spp., Multiplex PCR, Identification, Specificity, Whole genome sequence comparison

Funding: Hubei Academy of Agricultural Sciences youth foundation 2023NKYJJ13 Natural Science Foundation of Hubei Province 2022CFB852 Key research and development plan of Hubei Province 2023BCB068 Experimental demonstration and model integration of green technology for prevention and control of soybean root rot in Hubei Province Key Technology Research and Demonstration Project of Hubei Agricultural Science and Technology Innovation Center 2020-620-000-002-07 This work was supported by the Hubei Academy of Agricultural Sciences youth foundation (2023NKYJJ13), the Natural Science Foundation of Hubei Province (2022CFB852), the Key research and development plan of Hubei Province (2023BCB068), the Experimental demonstration and model integration of green technology for prevention and control of soybean root rot in Hubei Province, and the Key Technology Research and Demonstration Project of Hubei Agricultural Science and Technology Innovation Center (2020-620-000-002-07). The funders had no role in study design, data collection and analysis, decision to publish, or preparation of the manuscript.

==============================
Fusarium crown rot (FCR), caused by Fusarium spp., is a devastating disease in wheat growing areas. Previous studies have shown that FCR is caused by co-infection of F. graminearum, F. pseudograminearum, F. proliferatum and F. verticillioides in Hubei Province, China. In this study, a method was developed to simultaneously detected DNAs of F. graminearum, F. pseudograminearum, F. proliferatum and F. verticillioides that can efficiently differentiate them. Whole genome sequence comparison of these four Fusarium spp. was performed and a 20 bp sequence was designed as an universal upstream primer. Specific downstream primers of each pathogen was also designed, which resulted in a 206, 482, 680, and 963 bp amplicon for each pathogen, respectively. Multiplex PCR specifically identified F. graminearum, F. pseudograminearum, F. proliferatum and F. verticillioides but not from other 46 pathogens, and the detection limit of target pathogens is about 100 pg/μl. Moreover, we accurately determined the FCR pathogen species in wheat samples using the optimized multiplex PCR method. These results demonstrate that the multiplex PCR method established in this study can efficiently and rapidly identify F. graminearum, F. pseudograminearum, F. proliferatum, and F. verticillioides, which should provide technical support for timely and targeted prevention and control of FCR.

Introduction

Fusarium spp. is a conditional pathogen widely existing in plants and humans (Ma et al., 2013; Zhu et al., 2022). Fusarium spp. usually infects the vascular system of plants, disrupting the transportation of water and nutrients to tissues, causing plants to wither and rot (Hou et al., 2022). In addition to its pathogenic effects on plants and humans, Fusarium spp. also produces mycotoxins, such as fumonisin, zearalenone and fusaric acid, which are harmful to human and livestock health (Bertero et al., 2018; Munkvold, 2017). Wheat is a gramineous plant widely grown throughout the world and is one of the most important food crops that contribute significantly to human civilization, providing 19% of the daily caloric and 21% of the protein requirements for humans (Braun, Atlin & Payne, 2010; Tadesse et al., 2019). Fusarium crown rot (FCR), a soil-borne disease, is one of the most serious cereal diseases that affects the entire plant growth period in cereal crops and causes serious yield and quality losses worldwide (Kazan & Gardiner, 2018; Lin et al., 2022; Xu et al., 2017). FCR has been observed in many arid and semiarid wheat growing regions of the world, including the Americas (Cook, 1968, 1980; Fernandez & Zentner, 2005; Mishra et al., 2006; Smiley et al., 2005), Australia (Akinsanmi et al., 2004; Burgess, Wearing & Toussoun, 1975), Africa (Gargouri et al., 2011; Kammoun et al., 2009), New Zealand (Cromey, Parkes & Fraser, 2006), the Middle East (Gebremariam et al., 2018; Hameed, Rana & Ali, 2012; Pouzeshimiab et al., 2016), and China (Li et al., 2012; Xu et al., 2015, 2018; Zhang et al., 2015). In recent years, damage caused by FCR has gradually worsened in the Huanghuai wheat region of China. In many wheat growing areas in Henan Province, yield loss caused by FCR is up to 30–50% (Wang et al., 2022). In some high incidence areas, FCR caused yield losses up to more than 70%, with an average annual yield reduction of 9–35% from 2008 to 2019 (Luan et al., 2022).

FCR is commonly caused by several Fusarium spp., including F. pseudograminearum, F. graminearum, F. culmorum, F. avenaceum, F. verticillioides, and F. proliferatum. (Agustí-Brisach et al., 2018; Li et al., 2012; Meng et al., 2019; Zhang et al., 2015). Previous studies have shown that F. pseudograminearum is the predominant species to cause FCR in wheat but often mixed infection with other Fusarium spp. (Li et al., 2012; Kazan & Gardiner, 2018; Zhang et al., 2023). However, Zhang et al. (2015) reported that F. graminearum is the dominant pathogen of FCR in Anhui, Jiangsu, Henan, Shandong, and Hebei provinces of China. Thus, as a disease complex, the predominant pathogen of FCR may differ due to sampling location and ecological environment of the field (Saremi, Ammarellou & Jafary, 2007). Rapid and accurate identification of FCR pathogen species is of great importance, which may provide timely targeted prevention and control of FCR.

With the rapid development of molecular biology techniques, many molecular detection methods for pathogens have been developed. Compared with traditional detection methods based on isolation, cultivation, and morphological observation as well as biochemical characteristics, molecular identification method can be more accurate and efficient in the identification of the pathogens. Previous reports have shown that soil-borne diseases are often caused by pathogen complexes. For example, Fusarium spp., Botryodiplodia theobromae and Armillaria spp. are important fungal groups associated with cassava root rot (Bandyopadhyay et al., 2006). F. boothii, F. graminearum and F. meridionale mixed infection led to maize crown and root rot (Lamprecht et al., 2011). F. graminearum species complex (FGSC), which includes at least 16 known species, is the major cause of Fusarium head blight (FHB) in many parts of the world (Del Ponte et al., 2014). F. oxysporum f. sp. melonis and Monosporascus cannonballus co-infection causes melon radicle necrosis and rot (Wu et al., 2021). Therefore, identification of a single pathogen cannot meet the requirements of disease complex identification. Compared with single PCR, multiplex PCR possesses higher detection efficiency and can detect several pathogens simultaneously, which might reduce cost and save time. In order to efficiently and accurately monitor the occurrence of FCR, it is critical to develop a detection method that can detect multiple Fusarium spp. simultaneously.

Multiplex PCR amplify multiple target sequences simultaneously and has been used for detecting DNA of pathogen in medicine, environmental and agricultural sciences (Ali et al., 2015; Asano et al., 2010; Rappo et al., 2016). Previous reports have shown that multiplex PCR has enabled simultaneous detection of F. oxysporum sp. lycopersici, Clavibacter michiganensis subsp. michiganensis, Leveillula taurica, and begomoviruses on tomato plants (Quintero-Vásquez et al., 2013). F. verticillioides, F. subglutinans, and other species of the Gibberella fujikuroi complex were identified by PCR assays in maize seeds (Faria et al., 2012). Multiplex PCR was also reported to specifically identify F. oxysporum, Sclerotium rolfsii, and Lasiodiplodia theobromae in Peanut (Wang et al., 2023) and Listeria monocytogenes serovars, Listeria spp., and other species based on the target genes LMxysn_1095, lmo1083 and smcL (Feng et al., 2020). In these previous reports, a primer pair is designed for each pathogen, resulting in an excessive number of primers in a multiplex PCR system. Too many primers processed simultaneously in a PCR system may lead to primer cross-binding and primer dimer formation, thus reducing amplification efficiency. Whole genome sequence comparison can be used to identify universal primers for multiple pathogens, thus reducing the total number of primers in a multiplex PCR molecular detection system, which is an easier and more efficient choice (Hu et al., 2021; Kim et al., 2015; Liu et al., 2024; Park et al., 2017; Yu et al., 2019).

Our previous investigation showed that FCR mainly consists of F. graminearum, F. pseudograminearum, F. proliferatum and F. verticillioides in wheat in Hubei Province, China. To develop a multiplex PCR system for detecting these four Fusarium spp., we designed primer sets based on the genome sequence of F. pseudograminearum Class 2-1C (GenBank accession number CP064755.1), F. graminearum PH-1 (GenBank accession number HG970332.2), F. proliferatum ET1 (GenBank accession number NW_022194799.1), and F. verticillioides 7600 (GenBank accession number CM000579.1). A 20 bp sequence was selected as a universal upstream primer and specific downstream primers of four Fusarium spp. with different amplicon size were designed. We then evaluated the specificity and applicability of this method in accurately detecting four Fusarium spp. in infected wheats.

Materials and Methods

Fungal strains, culture conditions, and DNA extraction

A total of 22 strains of F. graminearum, F. pseudograminearum, F. proliferatum and F. verticillioides were collected by Hubei Academy of Agricultural Sciences, and a total of 46 fungal strains, e.g., Fusarium solani, Fusarium incarnatum, Fusarium equiseti, Fusarium oxysporum, Fusarium humuli, Fusarium brachygibbosum, Fusarium fujikuroi were kindly provided by Nanjing Agricultural University, Jiangsu Academy of Agricultural Sciences, Northwest Agriculture and Forestry University, and Yulin Normal University of China. Fungal strains culture and DNA extraction method were as previously described in Liu et al. (2024). All strains were routinely cultured on potato dextrose agar (PDA) plates (200 gL−1 of potato extracts, 1% glucose, and 2% agar), and incubated at 25 °C culture for 7–10 days. Mycelia of each isolate were collected with a sterile spatula for DNA extraction. Genomic DNA was extracted from mycelia using the Plant Genomic DNA Kit DP305 (TIANGEN, Beijing, China) according to the manufacturer’s instructions. DNA samples were measured with spectrophotometry to determine quality and concentration and stored at −20 °C until use.

Comparative genomics for identifying multiplex PCR primers

The genome sequences of F. pseudograminearum Class 2-1C (GenBank accession number CP064755.1), F. graminearum PH-1 (GenBank accession number HG970332.2), F. proliferatum ET1 (GenBank accession number NW_022194799.1), F. verticillioides 7600 (GenBank accession number CM000579.1), F. gerlachii CBS 119176 (GenBank accession number GCA_017656835.1), F. boothii CBS 316.73 (GenBank accession number GCA_017656985.1), F. culmorum Class2-1B (GenBank accession number CP064747.1), F. aethiopicum CBS 122858 (GenBank accession number GCA_017657045.1) and F. vorosii CBS 119178 (GenBank accession number GCA_017656575.1) were downloaded from the National Center for Biotechnology Information (NCBI) database. The primer sets design method were as previously described in Liu et al. (2024). We performed multiple alignments of the conserved sequences using Mauve software (version 2.3.1) to obtain homologous gene sequence fragments of these genomes. The same ≥20 bp sequences were selected from the homologous fragments of these genomes. Then specific downstream primers were designed from downstream non-homologous sequences. Therefore, a ≥20 bp genome sequence was selected from homologous fragments in F. pseudograminearum, F. graminearum, F. proliferatum, and F. verticillioides, and served as a universal upstream primer. A 1,000 bp downstream sequence was obtained in each genome for sequence alignment using BioEdit software (version 7.0.9.0). Then, nucleotide sequence of the designed specific downstream primers of each target strain was verified in the Basic Local Alignment Search Tool (BLAST) of the NCBI database. The primers are described in Table 1. The primer sets were synthesized by Sangon Biotech (Shanghai, China).

Table 1 Primers used in the multiplex PCR.

Target organisms	Primer	Primer sequence (5′-3′)	Length of production	
Fusarium spp.	Fu-4F	CTTGAACCTGAGACCTTCGC		
Fusarium graminearum	Fgram-R	CTCATAGCGATATTCTCGTATAC	206 bp	
Fusarium pseudograminearum	Fpseu-R	CGCACATTGCTTATTGCTTA	482 bp	
Fusarium proliferatum	Fprol-R	ATTCACGGATGAGAATCAAG	680 bp	
Fusarium verticillioides	Fvert-R	TCAAAGGAATGTCCGGTAGA	963 bp	

Optimization of multiplex PCR condition for detection of four Fusarium spp.

Multiplex PCR assay-related parameters were evaluated and optimized, including primer annealing temperatures, primer, dNTPs and Mg2+ concentrations. The test method were as previously described in Liu et al. (2024). Multiplex PCR was performed in 50 μl reaction volumes containing 0.25 μl TaKaRa Ex Taq polymerase (5 U/μl), 5 μl 10 × Ex Taq buffer (Mg2+-free), 1–8 μl (0.5, 1, 1.5, 2, 2.5, 3, 3.5, 4 mM) of MgCl2 (25 mM), 2–16 μl (0.1, 0.2, 0.3, 0.4, 0.5, 0.6, 0.7, 0.8 mM) of dNTPs mixture (2.5 mM each), and 1μl for each of the four fungal DNA templates (each DNA concentration: 1 ng/μl). Primer length and G+C content were important factors that influence the amplification efficiency of multiplex PCR. Therefore, to determine primer sets, primers with different lengths and G+C contents were designed in previous assays, and then the optimal primer set was verified by an annealing temperature gradient experiment. To adjust optimal concentration of each primer in the multiplex PCR system, different primer concentration combinations were tested, including four groups of concentration ratios for the universal upstream primers (Fu-4F) and downstream primers (Fgram-R, Fpseu-R, Fprol-R, and Fvert-R) (group I: 1:1; group II: 2:1; group III: 3:1; and group IV: 4:1). The final concentrations of each specific downstream primer were set at 0.05 μmol/L, 0.1 μmol/L, 0.15 μmol/L, and 0.2 μmol/L, respectively (Table S1). Multiplex PCR amplification was performed with the following program: 95 °C for 5 min, 32 cycles of denaturation at 95 °C for 30 s, annealing at 45–65 °C for 30 s, extension at 72 °C for 1 min and final extension for 10 min at 72 °C. Twelve temperature gradients were set, including 45, 46.1, 47.7, 50.5, 53, 55, 57.2, 59.4, 61.6, 63.4, 64.6 and 65 °C to determine the optimal reaction conditions for annealing temperature. PCR products were visualized under UV light after being size-fractionated by electrophoresis through a 2% agarose gel made with TAE buffer and stained with ethidium bromide solution. These experiments were repeated three times.

Multiplex PCR specificity test

The specificity test method was done as previously described in Liu et al. (2024). To evaluate the specificity of the multiplex PCR primer set, 1 μl of 22 target pathogen DNA (six F. graminearum, eight F. pseudograminearum, five F. proliferatum and three F. verticillioides) from different hosts and other 46 fungal strains were used as templates for multiplex PCR amplification under the optimized multiplex PCR system and conditions. In addition, in order to demonstrate DNA was present for each sample, the internal transcriptional spacer (ITS) segments of 46 fungal pathogens were amplified with ITS4/ITS5 primers (White et al., 1990). All strains are listed in Table 2. PCR products were visualized under UV light after being size-fractionated by electrophoresis through a 2% agarose gel made with TAE buffer and stained with ethidium bromide solution. This experiment was repeated three times.

Table 2 List of fungal strains used in study.

Serial number	Strainsa	Host species	Sourceb	Amplification resultc	
	Target pathogens				
1	Fusarium graminearum*	Wheat	HBAAS	+	
2	Fusarium graminearum*	Wheat	HBAAS	+	
3	Fusarium graminearum*	Maize	HBAAS	+	
4	Fusarium graminearum*	Maize	HBAAS	+	
5	Fusarium graminearum*	Maize	HBAAS	+	
6	Fusarium graminearum*	Rice	HBAAS	+	
7	Fusarium pseudograminearum*	Wheat	HBAAS	+	
8	Fusarium pseudograminearum*	Wheat	HBAAS	+	
9	Fusarium pseudograminearum*	Wheat	HBAAS	+	
10	Fusarium pseudograminearum*	Wheat	HBAAS	+	
11	Fusarium pseudograminearum*	Wheat	HBAAS	+	
12	Fusarium pseudograminearum*	Maize	HBAAS	+	
13	Fusarium pseudograminearum*	Maize	HBAAS	+	
14	Fusarium pseudograminearum*	Soil	HBAAS	+	
15	Fusarium proliferatum*	Wheat	HBAAS	+	
16	Fusarium proliferatum*	Wheat	HBAAS	+	
17	Fusarium proliferatum*	Maize	HBAAS	+	
18	Fusarium proliferatum*	Soil	HBAAS	+	
19	Fusarium proliferatum*	Soil	HBAAS	+	
20	Fusarium verticillioides*	Wheat	HBAAS	+	
21	Fusarium verticillioides*	Wheat	HBAAS	+	
22	Fusarium verticillioides*	Maize	HBAAS	+	
	Other pathogens				
1	Fusarium solani	Tomato	HBAAS	−	
2	Fusarium incarnatum	Tomato	HBAAS	−	
3	Fusarium equiseti	Pepper	HBAAS	−	
4	Fusarium oxysporum	Wheat	HBAAS	−	
5	Fusarium oxysporum	Tomato	YLNU	−	
6	Fusarium oxysporum	Pepper	HBAAS	−	
7	Fusarium oxysporum	Watermelon	NJAU	−	
8	Fusarium oxysporum	Tobacco	NJAU	−	
9	Fusarium oxysporum	Cucumber	NJAU	−	
10	Fusarium humuli	Tomato	HBAAS	−	
11	Fusarium brachygibbosum	Tomato	HBAAS	−	
12	Fusarium fujikuroi	Rice	HBAAS	−	
13	Alternaria alternata	Tomato	HBAAS	−	
14	Alternaria spp	Liriodendron chinese	NJAU	−	
15	Ascochyta pisi Libert	Pea	NJAU	−	
16	Botryophaeria dothidea	Peach	NJAU	−	
17	Botrytis cinerea	Strawberry	NJAU	−	
18	Botrytis cinerea	Cucumber	NJAU	−	
19	Cercospora kikuchii	Soybean	NJAU	−	
20	Colletorichum lagenerium	Watermelon	NJAU	−	
21	Colletotrichum gloeosporioides	Pear	NJAU	−	
22	Diaporthe phaseolorum	Soybean	NJAU	−	
23	Glomerella cingulata	Tea	NJAU	−	
24	Leptosphaeria biglobosa	Oilseed rape	NJAU	−	
25	Leptosphaeria maculans	Oilseed rape	NJAU	−	
26	Mycosphaerella melonis	Watermelon	NJAU	−	
27	Mycosphaerella melonis	Cucumber	NJAU	−	
28	Ophiostoma ulmi	Elm	NJAU	−	
29	Pestalotiopsis theae	Tea	NJAU	−	
30	Phellinidium lsulphurascens	Pine	NJAU	−	
31	Phialophora gregata	Soybean	NJAU	−	
32	Phoma pinodella	Pea	NJAU	−	
33	Phoma spp	Jujube	NJAU	−	
34	Phomopsis amygdali	Peach	NJAU	−	
35	Phomopsis fukushii	Pear	NJAU	−	
36	Phomopsis helianthi	Sunflower	NJAU	−	
37	Phomopsis longicolla	Soybean	NJAU	−	
38	Phomopsis truncicola	Apple	NJAU	−	
39	Rhizoctonia cerealis	Wheat	JAAS	−	
40	Rhizopus oryzae	Soil	NJAU	−	
41	Sclerotinia sclerotiorum	Cauliflower	HBAAS	−	
42	Sclerotium rolfsii	Pepper	HBAAS	−	
43	Stenocarpella maydis	Maize	NJAU	−	
44	Verticillium albo-atrum	Alfalfa	NJAU	−	
45	Verticillium dahliae	Tomato	NWAFU	−	
46	Verticillium dahliae	Wheat	NJAU	−	
Notes:

a Asterisks (*) indicate the target pathogens.

b HBBAS, Hubei Academy of Agricultural Sciences; JAAS, Jiangsu Academy of Agricultural Sciences; NWAFU, Northwest Agriculture and Forestry University; NJAU, Nanjing Agricultural University; YLNU, Yulin Normal University; HBAAS, Hubei Academy of Agricultural Sciences.

c Specificity test results of multiplex PCR are indicated as positive (+) or negative (−).

Multiplex PCR sensitivity test

The sensitivity test method was done as previously described in Liu et al. (2024). To determine the sensitivity of the multiplex PCR assay, genomic DNA from the four target pathogens was serially diluted to 10 ng/μl, 1 ng/μl, 100 pg/μl, 10 pg/μl, 1 pg/μl, 100 fg/μl, and 10 fg/μl by a 10-fold gradient with sterile double distilled water 1 μl of each DNA dilution concentration was used as a single PCR template to test the detection limit of each target pathogen by single PCR. Subsequently, each DNA dilution concentration was mixed, respectively, as a multiplex PCR template to test the detection limit of multiplex PCR for each target pathogen. PCR was performed according to the optimized conditions. Finally, PCR products were visualized under UV light after being size-fractionated by electrophoresis through a 2% agarose gel made with TAE buffer and stained with ethidium bromide solution. This experiment was repeated three times.

Detection of target pathogen DNA from field wheat samples and artificially inoculated wheat samples

To evaluate the applicability of the multiplex PCR assay for four Fusarium pathogens of FCR, we collected 22 wheat samples in a wheat growing area of Xiangyang (32.2015913°N, 110.901005°E) and Suizhou (31.9938899°N, 113.0270585°E) in Hubei Province of China in June 2022. DNA extraction method was done as previously described in Liu et al. (2024). After a small piece of tissue was excised from the stem of the 22 wheat samples using a sterilized scalpel, genomic DNA was extracted from field wheat samples using the Plant Genomic DNA Kit (TIANGEN, Beijing, China) according to the manufacturer’s instructions.

Artificial inoculation method was done as previously described in Liu et al. (2024). For the artificial inoculation test, fungal strains were cultured on PDA for three days at 25 °C, then mycelium plugs were transferred to mung bean medium and cultured at 25 °C for seven days with shaking at 200 rpm. Conidial suspensions were filtered through four layers of cheesecloth to separate conidia from mycelia. Concentration of the conidial spore suspensions was estimated using a hemocytometer and adjusted to 1 × 107 spores/ml. Wheats were inoculated with conidia suspensions of each fungus (1 × 107 spores/ml) in the stem of each wheat. Genome DNA was extracted after inoculation of 17 healthy wheats with four Fusarium strains in different combinations and seven wheats with sterile water for 3 days. Wheat samples inoculated with four Fusarium strains served as positive controls, while samples treated with sterile water were used as negative controls. Genomic DNA from all wheat samples was extracted using the Plant Genomic DNA Kit (TIANGEN, Beijing, China) according to the manufacturer’s instructions. All DNA extracted from the wheat sample used as a template for the multiplex PCR, which was performed using an optimized multiplex PCR system. PCR products were visualized under UV light after being size-fractionated by electrophoresis through a 2% agarose gel made with TAE buffer and stained with ethidium bromide solution. Amplified products of multiplex PCR were verified by sequencing of Sangon Biotech (Shanghai, China).

Results

Specific primers for four Fusarium spp. were designed via whole genome sequence comparison

To detect DNA from F. graminearum, F. pseudograminearum, F. proliferatum and F. verticillioides simultaneously, we screened specific primer combinations and established a multiplex PCR system (Fig. 1A). First, whole genome sequence comparison analysis identified a 20 bp sequence located within a tRNA-lle gene in the genomes of four Fusarium strains. This 20 bp sequence is located at nucleotide positions 1,558,947 to 1,558,966, 1,532,724 to 1,532,743, 2,012,079 to 2,012,098, and 2,435,141 to 2,435,160 in F. pseudograminearum Class 2-1C (GenBank accession number CP064755.1), F. graminearum PH-1 (GenBank accession number HG970332.2), F. proliferatum ET1 (GenBank accession number NW_022194799.1), F. verticillioides 7,600 (GenBank accession number CM000579.1) genome respectively (Fig. 1B). This sequence was selected as an upstream universal primer (Fu-4F), and specific downstream primers (Fgram-R, Fpseu-R, Fprol-R, and Fvert-R) of four pathogens with different amplicon sizes were designed. The amplicon size of F. graminearum, F. pseudograminearum, F. proliferatum and F. verticillioides were 206, 482, 680 and 963 bp, respectively (Fig. 1 and Table 1). In addition, the downstream primers matched only the sequence of the target pathogens.

Figure 1 Schematic design and location of primers for multiplex PCR detection of four Fusarium strains.

(A) The diagram represents the genomics sequences used to design the primers based on comparative genomics. Arrows indicate the positions and directions of the primers. (B) The genomic regions of F. pseudograminearum, F. graminearum, F. verticillioides and F. proliferatum used to design the universal upstream primer and specific downstream primers. Homologous bases are shaded in black. Each designed primer was marked with red rectangle. Fu-4F: Universal upstream primer. Fgram-R, Fpseu-R, Fprol-R, and Fvert-R: Specific downstream primers. Arrows indicate the positions and directions of the primers.

Standardization of for the multiplex PCR system

The length of the primers and the G+C content were closely related to the annealing temperature of PCR. The optimal primer set was selected by testing the annealing temperature of different primer sets (Fig. S1). Then, we tested the effects of different primer dNTPs, and Mg2+ concentration on the efficiency of multiplex PCR amplified DNA from the target pathogens. Our results showed that more PCR product was amplified under following primer concentrations (Fu-4F: 0.8 μmol/L, Fgram-R: 0.2 μmol/L, Fpseu-R: 0.2 μmol/L, Fprol-R: 0.2 μmol/L, Fvert-R: 0.2 μmol/L) when 2 mM MgCl2 and 0.2 mM dNTPs were added with 53 °C annealing temperature (Fig. 2).

Figure 2 Multiplex PCR amplification at different PCR reagent composition and conditions.

(A) Primer concentration ratio between the common forward primer Fu-4F and the specific reverse primer: 1:1 (group I), 2:1 (group II), 3:1 (group III) and 4:1 (group IV). Lane M: 2,000 bp DNA ladder, Lanes 1–4: concentration of each primer in group I, Lanes 5–8: . concentration of each primer in group II, Lanes 9–12: concentration of each primer in group III, Lanes 13–16: concentration of each primer in group IV. (B) MgCl2 concentrations. Lane M: 2,000 bp DNA ladder, lanes 1–8: 0.5, 1, 1.5, 2, 2.5, 3, 3.5, 4 mM, respectively. (C) dNTP concentrations. Lane M: 2,000 bp DNA ladder, lanes 1–8: 0.1, 0.2, 0.3, 0.4, 0.5, 0.6, 0.7, 0.8 mM, respectively. (D) Gradients of annealing temperature. Lane M: 2,000 bp DNA ladder, Lanes 1–12: 45, 46.1, 47.7, 50.5, 53, 55, 57.2, 59.4, 61.6, 63.4, 64.6 and 65 °C. Red rectangle indicates the optimal reaction system and conditions of multiplex PCR.

DNA from four target pathogens were specifically and sensitively detected by multiplex PCR

Using the optimal multiplex PCR system, an unambiguous detection result was obtained by multiplex PCR using mixed or individual genomic DNA of F. graminearum, F. pseudograminearum, F. proliferatum and F. verticillioides as templates. This result indicates that the established multiplex PCR method could specifically detect DNA of 22 target strains from different hosts (Fig. 3 and Table 2). As expected, DNA from other 46 fungal pathogens had no amplified product (Table 2 and Fig. S2). In addition, the fragments of the internal transcribed spacer (ITS) of 46 fungal pathogens could be amplified with the ITS4/ITS5 primer, demonstrating the presence of DNA in each sample (Fig. S3). Under the optimized PCR reaction conditions, no fragments were amplified without the addition of DNA templates, indicating that the fragments amplified by the primer set were due to the target DNA (Fig. S4).

Figure 3 Specificity of multiplex PCR.

Multiplex PCR primer sets only amplified the DNAs for F. graminearum, F. pseudograminearum, F. proliferatum and F. verticillioides. M: DL2000 marker; Mix: mixed DNA samples from the four Fusarium species; Lanes 1–3: F. verticillioides. Lanes 4–8: F. proliferatum. Lanes 9–16: F. pseudograminearum. Lanes 17–22: F. graminearum.

In addition, the results of the sensitivity test in three replicate tests show that the PCR detection limit for individual DNA was 10 pg/μl for F. verticillioides, 1 pg/μl for F. proliferatum, 100 pg/μl for F. pseudograminearum, and 100 pg/μl for F. graminearum (Figs. 4B–4E). However, the detection limit for multiplex PCR was about 100 pg/μl for DNA mixture from F. verticillioides, F. proliferatum, F. pseudograminearum, and F. graminearum (Fig. 4A).

Figure 4 Sensitivity of multiplex and single PCR assay.

(A) Sensitivity of multiplex PCR assay for F. graminearum, F. pseudograminearum, F. proliferatum and F. verticillioides at 100 pg/µL. (B) Sensitivity of PCR assay with Fu-4F/Fgram-R primer for F. graminearum at 10 pg/µL. (C) Sensitivity of PCR assay with Fu-4F/Fpseu-R primer for F. pseudograminearum at 1 pg/µL. (D) Sensitivity of PCR assay with Fu-4F/Fprol-R primer for F. proliferatum at 100 pg/µL. (E) Sensitivity of PCR assay with Fu-4F/Fvert-R primer for F. verticillioides at 100 pg/µL. Lane M: 2,000 bp DNA ladder, Lanes 1–7: 10 ng/μl, 1 ng/μl, 100 pg/μl, 10 pg/μl, 1 pg/μl, 100 fg/μl, and 10 fg/μl pure genomic DNA.

Multiplex PCR was successfully applied to detect pathogen DNA within wheat samples from the field and artificially inoculated samples

To determine the applicability of this multiplex PCR assay, we detected pathogen DNA in 22 wheat samples from the field and 24 artificially inoculated wheat samples. Our results showed that F. pseudograminearum, F. graminearum, and F. verticillioides were identified in 15, 10, and 3, respectively, of the 22 wheat samples from the field (Fig. 5A and Table S2). Among them, 10 wheat samples were co-infected with F. pseudograminearum and F. graminearum, two wheat samples were co-infected with F. pseudograminearum and F. verticillioides, and one wheat sample was co-infected with F. pseudograminearum, F. graminearum, and F. verticillioides (Fig. 5A). In addition, the results of the 24 artificially inoculated wheat samples were consistent as expected, F. pseudograminearum, F. graminearum, F. proliferatum, and F. verticillioides were identified in 12, eight, four, and four of the 24 artificially inoculated wheat samples, respectively (Fig. 5B and Table S2). Among, 10, 11, 14, 15, 16, 20, and 21 lanes were wheat samples inoculated with sterile water, and no amplification bands were found (Fig. 5B and Table S2). The amplified products were further verified by sequencing.

Figure 5 Multiplex PCR detection of four Fusarium strains in wheat samples from the field and artificially inoculated samples.

(A) Target pathogen strains were detected in wheat samples from the field at Xiangyang and Suizhou, Hubei province using multiplex PCR assay. Lane M: 2,000 bp DNA ladder; PC, positive control; NC, negative control, Lane 1–22: wheat samples. (B) Target pathogen strains were detected in artificially inoculated wheat samples using multiplex PCR assay. Lane M: 2,000 bp DNA ladder, Lanes 1–24: wheat samples. where lanes 10, 11, 14, 15, 16, 20, and 21: wheat samples were inoculated with sterile water.

Discussion

FCR is a common wheat disease caused by several Fusarium spp. (Meng et al., 2019; Scherm et al., 2013; Tunali et al., 2012). Due to different ecological environments in different regions, the composition of FCR-causing pathogens may be different. This makes phenotype-based and single PCR identification methods for FCR pathogen detection tedious and time-consuming. In this study, we designed a primer set via whole genome sequence comparison and developed a multiplex PCR assay to simultaneously detect DNA from four Fusarium species. This method will reduce the cost and time of pathogen analysis.

Multiplex PCR molecular detection methods have been applied to pathogen detection in medicine, environment, agricultural science, and other related fields (Ali et al., 2015; Asano et al., 2010; Rappo et al., 2016). Previously, multiplex PCR was reported to specifically identify Fusarium spp. Rhizoctonia cerealis, and Bipolaris sorokiniana based on ITS and TEF1-α in winter wheat (Sun et al., 2020). Previous reports have showed that multiplex PCR molecular detection method has enabled the simultaneous detection of DNA from F. oxysporum sp. lycopersici, C. michiganensis subsp. michiganensis, L. taurica, and begomoviruses on tomato plants (Quintero-Vásquez et al., 2013). F. verticillioides, F. subglutinans, and other species of the G. fujikuroi complex were also identified by PCR assays (Faria et al., 2012). Multiplex PCR was reported to specifically identify F. oxysporum, S. rolfsii, and L. theobromae in Peanut (Wang et al., 2023). A multiplex method RT-PCR based on five primer pairs was developed for differentiation and simultaneous diagnosis of five Porcine astroviruses (Liu et al., 2021). Usually, multiplex PCR contains numerous primers, leading to primer cross binding and primer dimer formation. In this study, we designed a single common upstream primer for simultaneous amplification of DNA from four Fusarium strains by reducing the number of primers in the PCR system.

With the rapid development of genome sequencing technology and bioinformatics, comparative genomics can be used to identify new molecular detection targets of pathogens and design universal upstream primers for multiple pathogens to reduce the number of primers (Hu et al., 2021; Kim et al., 2015; Liu et al., 2024; Park et al., 2017; Yu et al., 2019). In this study, we identified a 20 bp sequence as a common upstream primer based on comparative genomics to reduce the complexity of primers, and then specific downstream primers of F. graminearum, F. pseudograminearum, F. proliferatum and F. verticillioides were designed sequentially with different sequence fragment sizes. However, PCR application with multiple pair of primers combinations is complicated. We found that the length and G+C content of the primers affected the amplification efficiency when designed a multiplex PCR system. Therefore, we continuously adjusted the length and G+C content of the primer combination and finally designed a primer combination that could stably amplify F. graminearum, F. pseudograminearum, F. proliferatum and F. verticillioides.

The composition of PCR reagents and PCR conditions are key factors that influence multiplex PCR amplification (Zhao, Yuan & Huang, 2007). In the process of multiplex PCR, different primers will compete for other reaction components to amplify target DNA, so it is necessary to optimize the concentration of primer combination in the reaction system to ensure simultaneous amplification of multiple targets (Markoulatos, Siafakas & Moncany, 2002). In this study, we optimized the primer concentration as well as dNTPs, MgCl2 and annealing temperature, which also affect multiplex PCR results (Markoulatos, Siafakas & Moncany, 2002; Zhao, Yuan & Huang, 2007). PCR systems with dNTPs at 0.2–0.4 mM are usually the most favorable for amplification, and amplification is rapidly inhibited above this value, while lower dNTP concentration (dNTPs at 0.1 mM) allows PCR amplification with reduced products (Markoulatos et al., 1999; Markoulatos, Siafakas & Moncany, 2002). In addition, optimization of Mg2+ is crucial as excessive Mg2+ concentration stabilizes DNA double strand and prevents complete denaturation of DNA, thus reducing amplification yield, while insufficient Mg2+ concentration would also reduce PCR product (Markoulatos, Siafakas & Moncany, 2002; Deng et al., 2023). In this study, we optimized our PCR system with 0.8 μmol/L Fu-4F, 0.2 μmol/L Fgram-R, 0.2 μmol/L Fpseu-R, 0.2 μmol/L Fprol-R, 0.2 μmol/L Fvert-R, 2 mM MgCl2, 0.2 mM dNTPs with the annealing temperature of 53 °C in a 50 μl reaction.

In addition, we also analyzed the specificity, sensitivity, and detection limit of wheat samples based on the optimized reaction system and conditions. This study showed that multiplex PCR only amplified DNA of the target strains with the expected amplicon size, indicating that the designed primer sets had high specificity for detection of the target pathogens. In addition, the detection limit of multiplex PCR for F. graminearum, F. pseudograminearum, and F. verticillioides and F. proliferatum was 100 pg/μl, which can meet the requirements of low DNA concentration. This study showed that the sensitivity of multiple PCR was lower than that of single PCR. This result indicates the complexity of the amplification of multiple PCR primer sets, which may be due to the competitive utilization of Mg2+ and dNTP by primer sets, causing a decrease in amplification efficiency, thereby reducing the detection sensitivity of multiplex PCR. Moreover, we successfully identified the presence of these Fusarium strains in the wheat samples from the field and the artificially inoculated wheat samples using the established multiplex PCR. After three days of artificial inoculation of four Fusarium spp. on wheat, DNA was extracted from wheat samples for multiplex PCR detection before there were no disease symptoms. The results showed that the pathogen DNA could be accurately amplified, indicating that the multiplex PCR system can detect fungal pathogens before the appearance of disease. This is very important for the early identification of disease and the timely initiation of disease prevention and control measures.

Conclusion

We developed primer sets for F. graminearum, F. pseudograminearum, F. proliferatum, and F. verticillioides via whole genome sequence comparison and established a multiplex PCR method for simultaneous identification of four Fusarium spp. in a single PCR. This ability to detect four target pathogens in a single reaction is more cost-effective and saves time. Multiplex PCR system can specifically identify four target pathogens, but not 46 other fungal pathogens, with the detection limit of four target pathogens at 100 pg/μl. In addition, we accurately identified FCR pathogen species in wheat samples using the optimized multiplex PCR method. Therefore, the multiplex PCR method described here is a useful tool for diagnosing FCR pathogen species.

Supplemental Information

Supplemental Information 1 Supplemental Figures and Tables.

Supplemental Information 2 Optimization of multiplex PCR condition for detection of four Fusarium spp.

Original electrophoretic images of all multiplex PCR assays.

Additional Information and Declarations

Competing Interests

Author Contributions

Data Availability

The authors declare that they have no competing interests.

Siyi Deng performed the experiments, analyzed the data, prepared figures and/or tables, and approved the final draft.

Wei Chang performed the experiments, prepared figures and/or tables, and approved the final draft.

Quanke Liu performed the experiments, authored or reviewed drafts of the article, and approved the final draft.

Youfu Zhao analyzed the data, authored or reviewed drafts of the article, and approved the final draft.

Jun Liu conceived and designed the experiments, analyzed the data, prepared figures and/or tables, authored or reviewed drafts of the article, and approved the final draft.

Hua Wang conceived and designed the experiments, prepared figures and/or tables, authored or reviewed drafts of the article, and approved the final draft.

The following information was supplied regarding data availability:

The raw measurements are available in the Supplemental Files 2. The raw data shows unprocessed electrophoresis gel images from PCR experiments.

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
