# Peer review of "Development and application of multiplex PCR for the rapid identification of four Fusarium spp. associated with Fusarium crown rot in wheat"

_PeerJ, doi:10.7717/peerj.17656_

## Round 0.1 · original submission · Minor Revisions

Dear Dr. Liu,
Please find enclosed the comments of three independent reviewers. Despite the minor recommendations, please pay attention to the validity of the findings pointed by reviewer 3:

(1) Include the results of "No Template Controls (NTC)" in figures 2, 3, 4 and 5
(2) Align the discussion section length and G+C content of the primer with the description in the Materials and Methods.
(3) Explain the differences of detection limits between individual and multiplex PCR.

Looking forward to receive the revised version,
Best regards,
Ana I. Ribeiro-Barros

·

Basic reporting

The manuscript titled “Development and application of multiplex PCR for
the rapid identification of four Fusarium spp. associated with Fusarium crown rot in wheat” is a well-written and devised study using multiplex PCR to determine presence/ absence of four Fusarium pathogens commonly found to contribute to Fusarium crown rot in wheat. The writing is clear and discovery notable. Grammar needs minor editorial revision in some areas noted below and on the annotated pdf file.

Basic Reporting:
All elements of the paper meet requirements. Figures are of high enough resolution, grammar only needs minor revision, introduction relevant, and sources cited appropriately.

Minor comments:

Line 114: An extra space is present in this sentence that needs to be removed.

Line 120: DNA is stated to be extracted using the Plant DNA Kit (TIANGEN). However, a kit with this name was not found to be produced by this company. There are related kit names though produced by TIANGEN such as the SMART Magnetic Plant DNA Kit. The authors need to specify exactly which kit was used for DNA extraction.

Line 13: The ‘is’ here is in present tense while the rest of the paragraph is in past tense. Please change to ‘was’.

Lines 132-134: Irregular spacing present between these lines which needs to be fixed.

Lines 146-147: 1 µl of fungal DNA was used in the PCR reactions. However, the authors did not state what concentration of DNA that 1 µl represented. This is essential information for reproducibility of the assay.

Line 236: It is unclear how many times the LoD assessment was reproduced. Acceptable values for LoD require serial dilutions to be repeated in triplicate with the lowest quantity amplified in 3/3 replicates which are often then further replicated an additional 20 times. Although the additional 20 replicates is not necessary unless doing specific assays, the lowest quantity amplified in 3/ 3 replicates should be clearly stated.

Line 248: In this sentence, the authors state “the results of the 24 artificially inoculated wheat samples were consistent as expected” but do not note what the expectation was. Please include this expectation for clarity.

Line 300: A space is missing between 53 and the degree symbol.

Lines 502-507: Spacing between these lines is awkward and needs to be corrected.

Line 502: Capital for “Primer” missing here.

Line 510: Missing space between number and degree symbol.

Line 514: Figure 3 is stated as having “mixed” DNA samples but only one band is evident for each sample. The assumption is that mixed DNA samples were tested in PCR using only one of each primer set but this is not clearly stated. Please clearly state what is meant by ‘mixed’ DNA for this assay.

Lines 516-521: Capitals are missing at the beginning of each sentence. Please fix.

Line 525: Capital for “Target pathogen..” is missing.

Experimental design

Experimental design is relevant and only missing validation of presence of amplifiable DNA in the 46 fungal species shown in figure S1 needs to be completed. This and the lack of notation regarding how many technical replicates were performed for each assay, especially the sensitivity assays, were the only observed weaknesses for methodology in the manuscript.

The research question was well defined and fills in an identified knowledge gap. All research conducted fits within the Aims and Scope of PeerJ.

Major Comments:

1. In Figure S1: Missing amplification of the 46 fungal species using a primer set known to amplify any and all fungi, such as ITS primers, to demonstrate amplifiable DNA was present for each sample tested.

Validity of the findings

Validity of data:
Conclusions were validated by results and demonstrated soundness of methodology.

Additional comments

Excellent manuscript with fairly minor revisions required.

Reviewer 2 ·

Basic reporting

The paper "Development and application of multiplex PCR for the rapid identification of four Fusarium spp. associated with Fusarium crown rot in wheat" aims to describe an end-point PCR method to detect four Fusarium species simultaneosly.
In my opinion the paper submitted is well written, easy to read and its structure is adequate .
I have some suggestion to improve this manuscript:
In introduction: in advantageous environmental conditions, the detected fungi are able to produce toxic secondary metabolites (For examples F.graminearum is able to syntetize DON, NIV). The mycotoxins problem should be discussed in the introduction.
In conclusion: Authors should emphasize the importance of recognizing fungal pathogens before damage becomes apparent. The early identification is important to activate actions to combat the disease.

Experimental design

In my opinion the experimental disign is appropriate. The experiments were well conducte. The specificity and sensitivity of the developed multiplex PCR have been properly tested.
I have only two questions about the used protocols:
Line 146: What is the concentration of fungal DNA used?
Line 194: After inoculation, how long did it take before the extraction was done?

Validity of the findings

In conclusion: Authors should emphasize the importance of recognizing fungal pathogens before damage becomes apparent. The early identification is important to activate actions to combat the disease.

Reviewer 3 ·

Basic reporting

I carefully revised the manuscript titled “Development and application of multiplex PCR for the rapid identification of four Fusarium spp. associated with Fusarium crown rot in wheat” that was submitted to the PeerJ Journal.
The manuscript focuses on the simultaneous identification by Multiplex PCR of the pathogens responsible for Fusarium crown rot (FCR), a severe and devasting wheat disease. FCR is a co-infection known to be caused by at least three fungal pathogens, F. graminearum, F. pseudograminearum, F. proliferatum, and F. verticillioides, which can act in different combinations. Multiplex PCR is a molecular tool that identifies multiple targets in a single PCR reaction. Therefore, multiplex PCR is relevant in plant pathology due to its cost-efficiency, low time consumption, and straightforward interpretation.

Experimental design

no comment

Validity of the findings

1) in the results section, no No Template Controls (NTC) are presented in fig 2, 3, 4 and 5
2) the discussion section length and G+C content of the primer” are framed in a way that is not indicated in the Materials and Methods.
3) It is not explained why the limits of detection for individual DNA are different ( and higher in some cases) from those of multiplex PCR.

Additional comments

Additionally, I’m providing the authors with a list of topics and corrections/suggestions that could enhance the manuscript. Please correct some type mistakes in the text, as in line 300.

List of topics and corrections/suggestions:
- Line 62 – Please indicate that Anhui, Jiangsu, Henan, Shandong, and Hebei are provinces of China
- Line 68-70 – indicat that “molecular identification method can be more accurate and efficient” in the identification of the pathogens.
- Line 70 – remove “however”
- Line 90 – after “PCR assays”, add “ in maize seeds”
- Line 107 – Please make it uniform along the text; upstream or forward, and downstream or reverse
- Line 113-115 – include the country
- Line 115 – clarify that the 46 fungal strains are/or not other Fusarium
- Line 122 – DNA integrity and RNA contamination is usually checked in agarose gel. Was this done?
- Line 131 – “is as” replace by “was as”
Lines 131-132: The phrase is not clear. Are the authors trying to say that the primers were designed in the conserved sequences? If so, please rewrite it to make it clear.
- Line 135 – “verticillioides,and “ replace with “ verticillioides, and”
- Line 143 – replace “dosage with “concentration”
- Line 143 – replace “Mg2+. . “ by “Mg2+. “
- Line 145 - replace “4 mM)of MgCl2 “with “ 4 mM) of MgCl2”
- Line 195 – include a reference to the Plant DNA kit used
- Line 153 – replace “is also” with “may be”
- Line 157 – not only the cost but the identification will be faster.
- Line 234 – replace replace 10ng by 10ng/ul (concentration).
- Line 262 – “ PCR was reported to specifically identify”. In what samples?
- Lines 262-267 – this is the same as in the introduction.
- Lines 273 – 273 – remove this phrase
- Line 282 – replace “of primer” with “with multiple pair of primers”
Line 283 - 286: How did the authors find that length and G+C content affect PCR efficiency? This part is not described in the MM.
- Line 306-307 – 100pg/ul, concentration

---

## Round 0.2 · Minor Revisions

Dear Dr. Liu,
Please incorporate the minor comments highlighted by the independent reviewers, highlighting the changes in the text so that I can quickly go through before the final acceptance.
Sincerely
Ana Ribeiro-Barros

·

Basic reporting

The language is clear and unambiguous. Only minor formatting issues were apparent. These are listed below.

Line 159: Change “are” to “were”
Line 180: Add the word “assays” after “designed in previous”
Line 194-195: Change to “These experiments were repeated three times.”
Line 207: Change to “This experiment was repeated three times.”
Line: 220: Change to “This experiment was repeated three times.”

Experimental design

All revisions for the experimental design were completed. The research question was well defined, study relevant, and methods appropriate.

Validity of the findings

The additional supplemental figures added needed support to conclusions. After these additions, I feel the findings are well validated.

Reviewer 2 ·

Basic reporting

In my opinion, the paper has been improved and can be accepted.

Experimental design

The experimental design is appropriate

Validity of the findings

In my opinion, the results are valid

Additional comments

No additional comments

Reviewer 3 ·

Basic reporting

Overall the authors answered all the questions and made the requested changes.

Experimental design

About my previous question concerning the absence of No Template Controls (NTC) in fig 2, 3, 4, and 5, the authors did not answer my question, maybe because  I did not explain myself well. I meant to say that the” blank” - all PCR reagents except DNA template - was missing. This is important because the absence of amplification in the “blank” is proof of amplification due to DNA. Therefore, the “blank” in the gel should have been presented in the corresponding gels.

Also concerning the Primer length and G+C, which was also a previous question, the authors did not completely answer my question, maybe because I did not explain myself well again. I think that the explanation in lines 186- 189 is irrelevant because the authors designed several primer sets (probably several for the same target) but in this paper, only the final ones were tested.

Validity of the findings

as previously

Additional comments

I still found some misspelling errors that should be corrected and give a list of some suggestions.
line 76 - should be " Hebei provinces of China"
line 163 - should be "Therefore, a ≥ 20 bp genome "
line 165 - should be "as a universal "
line 180-181 - better as " primer, dNTPs and Mg2+ concentrations. "
line 206 - should be " The specificity test method was done as previously described in Liu "
Lines 211-212 - should be "each sample, the internal transcriptional spacer (ITS) "
line 220 - should be " The sensitivity test method was done as previously described in Liu (2024). "
Lines 237-238 - should be "DNA extraction method was done as previously described in Liu (2024). "
line 242 - should be "Artificial inoculation method was done as previously described in Liu (2024). "
Lines 284-285 - better as " We tested the effects of different primer c, dNTPs, and Mg2+ concentration and annealing temperature combinations "
line 368 - In the discussion, the authors indicate a primer concentration optimization, but do not indicate the value. For the other PCR reagents, this is done.
line 387 - remove "concentration" after 100 pg/μl

---

## Round 0.3 · accepted · Accept

Dear Dr. Liu,
Thank you for submitting the revised version of the MS. This version incorporates all changes suggested by the independent reviewers, but one: in Line 160, use "WERE" not "ARE" (as indicated by the reviewer), i.e. "...specific downstream primers WERE designed from downstream non-homologous". Please ensure that this minor issue is solved at the proof stage.

Sincerely,
Ama I. Ribeiro-Barros